# Novel Gene-Correction-Based Therapeutic Modalities for Monogenic Liver Disorders

**DOI:** 10.3390/bioengineering9080392

**Published:** 2022-08-15

**Authors:** Mahsa Ghasemzad, Mahdieh Hashemi, Zohre Miri Lavasani, Nikoo Hossein-khannazer, Haleh Bakhshandeh, Roberto Gramignoli, Hani Keshavarz Alikhani, Mustapha Najimi, Saman Nikeghbalian, Massoud Vosough

**Affiliations:** 1Department of Regenerative Medicine, Cell Science Research Center, Royan Institute for Stem Cell Biology and Technology, Academic Center for Education, Culture and Research, Tehran 1665666311, Iran; 2Department of Molecular Cell Biology-Genetics, Faculty of Basic Sciences and Advanced Technologies in Biology, University of Science and Culture, Tehran 13145-871, Iran; 3Gastroenterology and Liver Diseases Research Center, Research Institute for Gastroenterology and Liver Diseases, Shahid Beheshti University of Medical Sciences, Tehran 1983969411, Iran; 4Nanobiotechnology Department, New Technologies Research Group, Pasteur Institute of Iran, Tehran 1316943551, Iran; 5Division of Pathology, Department of Laboratory Medicine, Karolinska Institute, 17177 Stockholm, Sweden; 6Laboratory of Pediatric Hepatology and Cell Therapy, Institute of Experimental and Clinical Research (IREC), UCLouvain, 1200 Brussels, Belgium; 7Department of Hepatobiliary Surgery, Abu-Ali-Sina Hospital, Shiraz University of Medical Sciences, Shiraz 71946-84334, Iran; 8Experimental Cancer Medicine, Institution for Laboratory Medicine, Karolinska Institute, 17177 Stockholm, Sweden

**Keywords:** monogenic liver disorders, gene therapy, gene-editing tools, viral/non-viral vectors

## Abstract

The majority of monogenic liver diseases are autosomal recessive disorders, with few being sex-related or co-dominant. Although orthotopic liver transplantation (LT) is currently the sole therapeutic option for end-stage patients, such an invasive surgical approach is severely restricted by the lack of donors and post-transplant complications, mainly associated with life-long immunosuppressive regimens. Therefore, the last decade has witnessed efforts for innovative cellular or gene-based therapeutic strategies. Gene therapy is a promising approach for treatment of many hereditary disorders, such as monogenic inborn errors. The liver is an organ characterized by unique features, making it an attractive target for in vivo and ex vivo gene transfer. The current genetic approaches for hereditary liver diseases are mediated by viral or non-viral vectors, with promising results generated by gene-editing tools, such as CRISPR-Cas9 technology. Despite massive progress in experimental gene-correction technologies, limitations in validated approaches for monogenic liver disorders have encouraged researchers to refine promising gene therapy protocols. Herein, we highlighted the most common monogenetic liver disorders, followed by proposed genetic engineering approaches, offered as promising therapeutic modalities.

## 1. Introduction

The liver is a key metabolic organ that carries out most of the critical and relevant biochemical functions and biosynthetic activities, mainly ascribed to parenchymal cells (also known as hepatocytes). Hundreds of hereditary liver diseases are monogenic disorders. A deficiency in one single protein, involved in critical metabolic function, may lead to congenital metabolic disorders. Due to a vast plethora of secreted proteins and transportation potential, the liver is an ideal target for gene therapy. Furthermore, the hepatic organ has a tropism toward genetic vectors, rendering this tissue an ideal target for current gene therapy technologies [1].

The majority of monogenic liver diseases are autosomal recessive, co-dominant, or sex-related, characterized by inborn errors of metabolism [1,2]. Solid organ (liver transplantation, LT) or cell (hepatocyte) transplantations have been validated and described as effective treatments to reverse such metabolic deficiency (Table 1). Both liver and hepatocyte transplantations suffer from the same limitations and impediments: a lack of enough donors and life-long immunosuppression to grant allo-acceptance [3,4,5,6]. Accordingly, new therapeutic modalities have been proposed and investigated, such as gene-replacement or gene-correction therapy [7].

Two main approaches have been described as effective to correct inborn gene errors in liver cells. In in vivo approaches, the gene of interest is transferred directly to the hepatocytes through appropriate vectors. Conversely, ex vivo methods require hepatic cells isolated from the patient’s organ, and, after gene transfer/editing, the hepatocytes are re-implanted into the patient’s liver [41] (Figure 1).

Viral vectors are important gene-transferring mediators. They are potentially administered locally, intra-hepatically, or vehiculated through the venous system [42]. Viral vectors used to conduct gene-based therapy to the liver include adeno-associated virus (AAV), lentivirus, retrovirus, and high-capacity adenoviral vectors [2]. Studies have demonstrated that recombinant AAV accumulated in the liver without reiterating infection in humans [1] and could be used to specifically suppress or induce genes and microRNAs in the hepatocytes [43]. Lentiviral vectors can provide stable gene expression in hepatocytes and reduce insertional mutations, generally caused by retroviral vectors [1,43]. Retroviral vectors have relatively large transgene capacities and have been applied in several ex vivo gene therapies [1]. Non-viral vectors are synthesized in laboratories and have fewer safety concerns and relatively easy-to-construct features compared to classical viral vectors. However, non-viral vectors have restricted ability to deliver long-lasting transgene expression [44]. Lipid nanoparticles targeting liver tissue are promising gene delivery vehicles. Extracellular vesicles containing target mRNA have been administered and described as non-immunogenic, non-toxic, and easy to use compared to viral vectors [45] (Figure 2).

Besides delivering vectors, genome editing tools, such as transcription-activator-like effector nucleases (TALENs) or the CRISPR-Cas9 system, are the core of gene-editing technology for the correction of genetic errors. TALENs are artificial nucleases that can target and cleave the DNA sequences in animal models [43]. The CRISPR-Cas9 system has demonstrated outstanding plasticity and efficacy, leading to promising gene-editing technology with high efficiency and simple manipulation [45]. However, a critical challenge in efficient delivery of the CRISPR-Cas9 system is the limited size of viral vector transgenes, particularly in AAV vectors [46]. Within the following pages, we will highlight the most common monogenic hereditary liver disorders and correlate them with gene-correction-based therapeutic modalities. Additionally, clinical studies related to gene-correction-based therapies will be discussed (Table 2).

## 2. Familial Hypercholesterolemia

Familial hypercholesterolemia (FH) is a common autosomal dominant disorder characterized by high serum levels of low-density lipoprotein-C (LDL-C) and frequently associated with cardiovascular disorders [47]. FH is caused by mutations in low-density lipoprotein receptor (*LDLR*), apolipoprotein B (*APOB*), or convertase subtilisin/kexin type 9 (*PCSK9*) genes [48]. Two subtypes of FH are known as homozygous (HoFH) and heterozygous FH (HeFH). Recent data have shown that the incidence of homozygotic FH is 1 in 1,000,000, while HeFH is 2000 times more frequent (1:500) [49]. LT is considered to be a gold standard treatment, but it is invasive and may cause other complications [50]. LDL apheresis is another therapeutic option, limited due to high costs and difficulty to apply [51]. Furthermore, administration of statins, such as atorvastatin, only reduced the plasma LDL-C level by 10 to 25% [49].

One inspiring preclinical study applied an ex vivo gene therapy approach to ameliorate FH in an LDLR-deficient rabbit model. The transduction of a functional *LDLR* gene resulted in a decreased level of serum cholesterol [52]. A few years later, Tomita et al. evaluated the therapeutic effect of an LDL receptor delivered by hemagglutinating virus of Japan (HVJ)-liposomes in an LDL knockout mouse model. HVJ-liposomes could be used as the feasible gene delivery tools as they require a shorter incubation time than other gene delivery methods and have demonstrated attenuated cytotoxicity. The authors showed a reduced total cholesterol level [53]. Kassim et al. investigated the effect of gene therapy on atherosclerosis in a mouse model with HF. Adeno-associated virus 8 (AAV8) vectors were injected in order to transfer the *LDLR* gene to *LDLR*^−/−^/*APOB*^−/−^ mice. An immuno-histochemical analysis showed regression of atherosclerosis lesions [51]. In addition, some studies revealed that AAV 8 vectors could reduce plasma cholesterol levels in HF patients [49]. In another study, the missing LDLR in familial hypercholesterolemia liver chimeric mice was replaced by AAV-9-based gene therapy and the normal lipoprotein profiles after administration of a single dose were restored. In this study, human metabolic disease is induced in an experimental animal model by human hepatocyte transplantation and treated by gene therapy [54]. Li et al. performed exosome-based *LDLR* gene therapy in an FH mouse model. Mice were treated with exosomes containing *LDLR* mRNA and fed with a high-fat diet for 8 weeks. The results showed that this intervention resulted in *LDLR* expression and decreased the atherosclerosis and LDL cholesterol levels [45]. Regulatory non-coding RNAs and synthetic RNAs are also used for the treatment of FH. For example, Mipomersen (Kynamro) is a second-generation 20-nucleotide ASO that binds to ApoB100 mRNA in the liver and thereby reduces the plasma LDL-C concentrations [55]. Inclisiran (ALN-PCSsc PCSK9), an siRNA that inhibits the PCSK9 mRNA, has a promising new therapeutic approach for LDL-C reduction [56]. In a study, induced pluripotent stem cells (iPSCs) from a homozygous LDLR-null FH-patient (FH-iPSCs) were generated. FH-iPSCs were genetically corrected using the CRISPR-Cas9 system and the targeted integration of a correction cassette at the AAVS1 locus. The genetic editing resulted in restoration of LDLR expression and function [57].

## 3. Gaucher Disease

Gaucher disease (GD) is the most common lysosomal storage inherited disorder by recessive mutations in the *GBA* gene. GD is caused by defective glucocerebrosidase activity, leading to accumulation of glucocerebroside in lysosomes in several organs, such as the spleen, liver, bone marrow, and bone cells [58]. Thereby, hepatosplenomegaly, cytopenia, neurologic disorders, and bone-related diseases may occur in GD patients [59]. The prevalence of GD in the Ashkenazi Jewish population is about 1 in 850, compared to 1–2 per 100,000 in the general population [60]. Enzyme replacement therapy (ERT), reduction in glucosylceramide synthesis using substrate inhibitors (substrate depletion therapy), and allogeneic bone marrow transplantation are some of the treatment modalities currently offered to GD patients. Such treatments have some disadvantages, such as high cost, elevated toxicity, poor efficiency, lack of proper donors, and frequent immunogenic reactions after transplantation [58]. Gene therapy has been described as a promising alternative approach. However, the lack of suitable animal models for GD have limited the therapeutic developments and their validation [61]. One preclinical study reported the delivery and expression of the human *GBA* gene into bone marrow cells by retroviral vectors. The results demonstrated that *GBA* transduction could retain activity [62]. In another study, Enquist and colleagues generated GD knockout mice by deleting *GBA* exons 9–11 and reported that gene therapy by retrovirus vectors could ameliorate the GD phenotype and normalized the GC activity [61]. Moreover, researchers transferred a rAAV vector containing *GBA* cDNA into the fibroblasts and increased its activity from 1.9 to 4.6. Intravenous administration of vectors in wild-type mice resulted in efficient transduction into the tissues, and the GC activities of the liver, spleen, and lung were increased significantly [58]. Massaro et al. systemically delivered an AAV serotype 9 carrying the human *GBA* gene under control of a neuron-specific promoter to an nGD mouse model. The results demonstrated an increase in the life span of treated animals, rescue in the lethal neurodegeneration, normalization in the locomotor behavioral defects, and amelioration in the visceral pathology [63]. Zhao et al. developed a human iPSC line (SMBCi004-A) from an 8-year-old female patient with Gaucher disease. As a gene therapy approach, some reprogramming factors, such as OCT4, SOX2, KLF4, and miR-302–367, were delivered using a non-integrative plasmid and the results showed complete pluripotency, normal genetic stability, and the ability to differentiate into three germ layers [64]. Moreover, Diaz-Font et al. used an siRNA for the inhibition of the GCS gene as a potential therapeutic strategy for GD [65].

## 4. Mucopolysaccharidosis

Mucopolysaccharidosis (MPS) is another lysosomal storage disease caused by accumulation of glycosaminoglycans (GAGs) in blood and other organs (such as respiratory system, liver, spleen, central nervous system, and bone marrow). There are seven types of MPS, classified based on different mutations [66]. MPSs are all autosomal recessive disorders, with the exception of MPS type II. American epidemiological studies identified the MPS prevalence rate in approximately 1:100,000 live births in the United States [67]. ERT and hematopoietic stem cell transplantation (HSCT) have been offered as treatments for MPS (muco3), limited by high cost and need for weekly injections. Such cell-based treatments have demonstrated limited penetration of the blood–brain barrier (BBB) [66]. Thereby, gene therapy has been recently proposed as novel therapeutic modality, with predicted higher penetrability through the BBB. An in vivo study showed that gene therapy using lentiviral vectors improved the effects of HSCT and also ameliorated the MPS I phenotype, such as neurological and skeletal manifestations. Moreover, the results demonstrated the partial clearance of GAGs from liver and kidneys [68]. Di Domenico and colleagues injected lentiviral L-iduronidase (IDUA) vector into a murine MPS I model, correcting the GAG levels in the liver, urine and spleen [69]. In another preclinical study, the therapeutic effects of AAV-8 vector carrying *GALNS* resulted in significant improvement in MPS IVA-deficient murine bone and heart [70]. Another gene-editing study used a ZFN-mediated approach to correct the *IDUA* gene in MPS I murine hepatocytes. The corrected copy of the missing functional gene was inserted into the albumin locus, instrumental for stable and sustained enzymatic expression. As results, sufficient enzyme activity and improvement in the MPS I phenotype were monitored [71]. In a clinical trial (NCT02702115), SB-318 as a therapeutic ZFN-mediated genome editing tool was delivered by AAV-derived vectors. SB-318 was intended to function by replacement of the corrective copy of the IDUA transgene into the genome of the subject’s own hepatocytes and was expected to provide permanent and liver-specific expression of iduronidase for the lifetime of an MPS I patient. Muenzer et al. investigated the effects of SB-913 as a new type of investigational treatment for MPS II. SB-913 is a tool to insert a normal copy of the IDS transgene into the liver cells via AAV2/6 vectors. The results showed reduced GAG accumulation with lifelong continuous endogenous production of IDS [72].

As previously mentioned, CRISPR-Cas9 technology has recently changed the way we can modify DNA and correct MPS mutations ex vivo. MPS I patients’ fibroblasts have been transfected (by lipofectamine) and *IDUA* activity ameliorated, while lysosomal aggregation decreased [73]. A similar approach led to insertion of α-l-iduronidase (IDUA) cDNA without promoter into albumin locus in parenchymal cells. The AAV8 vectors containing the proprietary gene-editing system were injected into neonatal and adult MPS I mice, generating significant increments of IDUA enzymatic activity in the brain [74]. In another study, using CRISPR-Cas9, the authors corrected murine CD34^+^ hematopoietic stem cells targeting the lysosomal enzyme iduronidase into the CCR5 locus, leading to iduronidase secretion and improved biochemical and phenotypic abnormalities in an MPS I model [75]. Similarly, CRISPR-Cas9 insertion into the AAVS1 locus generated a 40% increment in GALNS activity, while lysosomal mass, total GAGs, and oxidative stress were normalized [76]. Miki et al. used the ex vivo gene-editing therapy using induced pluripotent stem cell (iPSC) and CRISPR/Cas9 technologies in an MPS Type 1 disease mouse model. After inducing fibroblast differentiation, the gene-corrected iPSC-derived fibroblasts demonstrated Idua function equivalent to the wild-type iPSC-derived fibroblasts [77].

## 5. Urea Cycle Defects

### 5.1. Ornithine Transcarbamylase Deficiency

Ornithine transcarbamylase (OTC) is a hepatic mitochondrial enzyme crucial in conversion of nitrogenous biomolecules into (excretable) urea [78]. OTC deficiency (OTCD) is a monogenic disease and the most common and severe defect of the urea cycle. OTC deficiency is an X-linked disorder with high frequency of new mutation rate and variable phenotypic consequences [79]. Urea cycle disorders affect 1 in 8200 US live births [80]. Recently, human-induced pluripotent stem cell line (SDQLCHi036-A) has been generated as a useful model to explore the pathogenesis and therapy-model platform of OTCD [81]. Although gene therapy could be a promising treatment for late-onset OTC deficiency, AAV gene therapy for neonatal cases offers only short-term therapeutic effects since non-integrated genomic material is lost during hepatocyte proliferation. Targeted mRNA therapy is another gene therapy approach for the treatment of OTCD. G. Prieve et al. designed a nanoparticle mRNA delivery system as a highly effective means of intracellular enzyme replacement therapy (i-ERT). The results demonstrated that nanoparticle mRNA delivery of human OTC mRNA normalizes plasma ammonia and urinary orotic acid levels [82]. Recently, CRISPR-Cas9 technology has been proposed as an effective gene-editing approach to correct a patient’s own cells, both in vivo and ex vivo. A recent study demonstrated homologous repair in 10% of the OTC alleles in the liver of newborn OTC spf^ash^ mice [83]. Recently, an effective and practical editing approach based on the CRISPR-Cas9 correction of OTC-deficient cells was reported, where a patient’s own hepatocytes were reprogrammed into iPS cells and later edited in order to correct OTC deficiency, generating genetically and phenotypically proficient hepatocytes [84]. The same group applied such correction ex vivo in a patient’s cells, where the selective deletion of a mutant intronic splicing site led to restoration of the urea cycle [3].

The successful CRISPR-Cas9 correction was also shown in the “humanized liver” OTC mice model, with OTC enzyme activity, enhanced clearance of ammonia, and reduced urinary orotic acid. Gene delivery systems with either an adeno-associated virus (AAV) or lipid nanoparticle containing mRNA have been proposed to be effective for the treatment of OTC deficiency [1,85]. ARCT-810 is a medicinal product containing OTC mRNA embedded in lipid nanoparticles (LNPs) [82]. DTX301 (scAAV8OTC) is a non-replicating and recombinant scAAV8 encoding human OTC and is currently undergoing safety evaluation and dose-finding tests [86]. In another study, the liver-tropic AAVLK03 gene transfer technology containing the *OTC* gene has been used for the treatment of cynomolgus monkeys. The results highlighted supra-physiological OTC overexpression with no adverse clinical events [87].

### 5.2. Citrullinemia Type I

Citrullinemia type I (CTLN I) is a urea cycle disorder with an autosomal recessive inheritance trait caused by a lack of arginosuccinate synthetase (ASS) enzyme activity encoded by mutated *AAS1* gene. The incidence rate is 1:250,000 individuals [88], and deficiency in ASS activity induces the aggregation of toxic metabolites, such as citrulline or ammonia, in patients’ plasma [89], resulting in many clinical manifestations, such as vomiting, inappetence, and lethargic status, right after birth [90]. Restriction in protein intake and supplementation with ammonia scavengers in the diet are palliative treatments, while LT, once again, is the sole established curative treatment [91]. Yukie et al. used disease-specific iPSCs for the modeling of the CTLN I. This model improved the understanding of CTLN I pathophysiology and could be used to pursue new therapeutic approaches [92]. Accordingly, cell and gene therapies have been investigated and proposed. Almost 30 years ago, Demarquoy reported the use of retroviral-mediated gene-correction on CTLN I patients’ fibroblasts to normalize *ASS1* expression [93]. Later, Patejunas and colleagues generated the first model for CTLN I (*ASS1 KO* mouse) and applied homologous recombination by electroporation of a vector containing *ASS1* gene fragment into embryonic stem cells [94]. In another study, the liver-targeted AAV8 vector for CTLN I in a murine model was investigated. The authors used a thyroxine binding globulin (TBG) promoter to induce expression of *ASS1* gene, generating reduced ammonia and plasma citrulline levels [16].

## 6. Alpha-1-Anti Trypsin Deficiency

Alpha-1-anti Trypsin deficiency (AATD) is one of the most common hereditary liver diseases, characterized by low serological levels of AAT [95]. AATD is caused by mutations in the *SERPINA1* gene. Polymerization of misfolded proteins and retention in endoplasmic reticulum of hepatocytes, caused by homozygous Piz alleles, could lead to decreased circulating AAT levels in the most severe patients [96,97]. Alpha-1-anti Trypsin is a circulating protease inhibitor, protecting lungs and connective tissue from human neutrophil elastase released by triggered neutrophils [98]. AATD therapeutic modalities require combined lung and liver transplantation. Recently, He et al. constructed a ferret model of AATD. The results of this study demonstrated that the AAT-knockout and PiZZ ferrets model the progressive pulmonary and liver disease and may serve as a platform for gene therapy [99]. Currently, augmentation therapy has been successfully applied to AATD lung emphysema but is ineffective for liver manifestations and poorly adherent to some patients’ genotypes [100,101]. Recently, gene augmentation therapy has demonstrated sustained AAT expression by using AAV vectors [102]. A weekly dose of siRNA or targeted antisense oligonucleotides AAT in the PiZ genotype has shown remarkable preclinical and clinical outcomes in the reduction of the AAT-Z phenotype (*AAT* gene allele Z) in the affected hepatocytes [103,104]. Allogenic hepatocyte transplantations have shown efficacy and pulmonary protection [105], but the aforementioned limitations encourage novel genetic approaches, reinvigorated by CRISPR-Cas9 potential. The gene-editing tool targeting *SERPINA1* in the PiZ mice model successfully reversed the phenotype of AATD in the liver [106]. Adenine base editing by CRISPR-Cas9 technology corrected the Z mutation in the patient’s iPSC-derived hepatocytes (iHeps). The results demonstrated that aberrant AAT accumulation was reduced [107]. Shen and co-authors showed that CRISPR-Cas9 gene-editing technology could impressively decrease AAT-Z liver expression and recover moderate levels of wild-type AAT-M (*AAT* gene allele M) in an AATD mouse model [108]. Intrapleural delivery of AAT-coding AAVrh.10 vector showed sustained expression of human AAT in mice [109]. Janosz and colleagues reported in vitro generation of AAT MΦ, enabling to engraft into the pulmonary microenvironment and convert into alveolar macrophages [110]. Recently, serotype 8 adeno-associated virus (AAV 8/AVL) has been presented as a second-generation gene therapy for AATD, encouraged by superior antiprotease protection even in an oxidative stress situation [111]. Another group evaluated and described the cytosine and adenine base editing for potential treatment of AATD [112]. The authors reported that treatment with lipid nanoparticles formulated with base editing reagents can generate a durable edition of *SERPINA1* in the liver, increased serological AAT levels, and improved liver histology [112].

## 7. Tyrosinemia Type I

Hereditary tyrosinemia type 1 (HT1) is a rare autosomal recessive metabolic disorder, associated with severe liver and kidney damage. The HT1 is caused by a defect in fumarylacetoacetate hydrolase (FAH), the last enzyme in the catabolic pathway of tyrosine [113]. The acute form of HT1 leads to an early onset and severe liver failure, whereas the chronic form emerges later and includes renal dysfunctions [114]. Accumulation of extra amounts of fumarylacetoacetate could lead to acute cell apoptosis and severe liver dysfunction [115]. Such a cytotoxic effect has also been shown to be helpful to grant a growth advantage to allogenic FAH-proficient hepatocytes [116]. Since 1992, NTBC/nitisinone, a selective drug working as an inhibitor of 4-Hydroxyphenylpyruvate dioxygenase, has been offered as a successful treatment in combination with a restricted diet, poor in tyrosine and phenylalanine amino acids [117,118]. Recently, it was demonstrated that a generic and bioequivalent version of NTBC, NITYR, and another brand of nitisinone (Orfadin) could offset the high costs of HT1 treatment [119]. Unfortunately, such pharmacological treatments do not protect patients from developing hepatocellular carcinoma, supporting new and improved cell and gene therapeutic approaches. In 2018, for the first time, VanLith and colleagues showed that ex vivo hepatocyte-directed gene-editing using CRISPR-Cas9 could be a curative therapy in HT1 [21]. Furthermore, Hickey and co-workers reported hepatocyte-directed ex vivo gene repair using a lentiviral vector to express FAH as a good therapeutic approach rather than whole organ transplantation for HT1 [120]. Zhang and colleagues used a two-AAV system based on CRISPR-Cas9 to enhance in vivo hepatocyte gene repair in a model of hereditary tyrosinemia, reporting efficient gene-correction in neonatal hepatocytes (approximately 10.8% of parenchymal cells corrected) but lower efficiency (approximately 1.6%) in adult mice [121]. mRNA-mediated protein replacement could be a promising gene therapy concept for the treatment of HT1. In a preclinical study, genetically engineered FAH^−/−^ mice were treated with FAH mRNA loaded into dendrimer lipid nanoparticles, and the results showed statistically equivalent levels of TBIL, ALT, and AST compared to wild-type C57BL/6 mice [122]. In another study, the CRISPR-Cas9 gene-editing system was used to correct the genetic defect in newborn HT1 rabbits [123]. In that study, both CRISPR-Cas9 and donor templates were delivered via AAV, leading to normal liver and kidney structures and functions. Unlike traditional CRISPR-Cas9-homology-directed repair, base editing can correct point mutations without supplying a DNA-repair template. Studies have shown that the efficiency of base editing could be improved with the inclusion of an N-terminal nuclear localization sequence and codon optimization for the Cas9 nickase [124,125]. More recently, Song and co-authors reported successful application of adenosine base editing to correct tyrosinemia with G•C to T•A point mutations [126].

## 8. Galactosemia

Galactosemia is a rare inherited metabolic disease that occurs due to mutations in genes involved in the galactose metabolism (Lelior pathway) [127]. GALT, galactosemia type I, is the most common and severe form of galactosemia, affecting 1:16,000–60,000 people worldwide, and may manifest early, when a newborn takes the first milk meal, leading to IQ and behavioral/physical disabilities [128]. The current treatment for galactosemia relies on galactose/lactose dietary restrictions, preventing the most severe forms of the disease [129]. To the best of our knowledge, any cellular or gene therapy approach has been tested or validated using the preclinical model of classic galactosemia (Galt KO mouse) [130,131]. Delivery of functional protein into the cells or enzymatic replacement therapies have been tested with limited success in lysosomal storage diseases [132,133], but any similar approach has been reported for GALT. Nevertheless, reports describing Galactokinase 1 (GALK1) induction, using the consensus reversal approach, resulted in a variety of proteins that maintained enzymatic activity and increased thermal stability [134]. Due to the required highly purified recombinant protein, enzymatic replacement therapy is an expensive approach that is safe by long-term recipients’ immune recognition [135]. Therefore, gene and mRNA therapies are emerging as promising medical therapies [136]. Balakrishnan and colleagues reported that multiple intravenous injections of human *GALT* mRNA into deficient mice were effective in inducing hepatic expression of mouse GalT protein and significantly reduced plasma galactose concentration level [137]. The same study showed that reduction and recovery of GALT activity may overcome galactose sensitivity in sick neonates [137]. Antisense technology has been shown to be effective in overcoming *GALT* splicing defects and restoring the splicing profile successfully [138].

## 9. Acute Intermittent Porphyria

Porphyria comprises a group of eight metabolic disorders characterized by defects in heme biosynthesis [139,140]. There are two major categories of hepatic porphyria: acute or inducible porphyria and chronic cutaneous porphyria [140]. Acute intermittent porphyria (AIP) arises from a defect in the hydroxymethylbilane synthase gene, encoding the third enzyme in the heme biosynthesis pathway [141], with a yearly incidence of 1.3:1,000,000 in Europe [142]. The early accumulation of heme precursors, such as d-aminolevulinic acid (ALA) and porphobilinogen (PBG), is associated with clinical features and, to a larger extent, the eventual development of hypertension, kidney failure, and liver cancer [143]. In the last decade, liver and domino liver transplantation have been used as treatment options for severely affected AIP individuals, characterized by strong and recurrent attacks [144]. Notably, clinical hepatocyte transplants have never been offered to AIP patients [116], while gene therapies based on siRNA-based platforms have been attempted and offered promising therapeutic effects. Three approaches are currently under clinical translation: (i) siRNA targeting ALAS1 transcript, aimed to reduce hepatic ALAS1 expression; (ii) by recombinant AAV-mediated transfer of human PBGD gene to enhance protein expression; (iii) administration of human PBGD mRNA packed in LPN [145]. In 2014, preclinical studies showed efficacy in liver-directed siRNAs targeting ALAS1 [146]. The first gene therapy trial for AIP and the first use of AAV5 in humans was reported by D’Avola and colleagues, proving the safety, but AIP metabolic correction was not achieved at the tested dose [147]. Givosiran (Givlaari™, Alnylam Pharmaceuticals, Cambridge, MA, USA) is an siRNA recently approved by both American and European regulatory agencies as a ribonucleic acid interference (RNAi) therapy that could target and downregulate ALSA1 transcription [148]. In another study, a recombinant protein formed by linking ApoAI to the amino terminus of human PBGD (rhApoAI-PBGD) has been used to transfer PBGD into liver cells, preventing AIP rise in a preclinical model [149].

## 10. Hemophilia

Hemophilia A and B are rare and recessive X-linked congenital diseases, caused by deficiency in coagulation factor VIII (FVIII) or IX (FIX), affecting one newborn every 5000 or 25,000 male births, respectively [150]. The hemophilic arthropathy pathophysiology is multifactorial and apparently caused by the interaction of blood with articular cartilage and driven by inflammation [151]. Patients with severe hemophilia experience recurrent bleeding events in joints, muscles, or soft tissues as results of traumatic events or with no apparent cause. Hemophilic patients may also suffer from life-threatening intracranial hemorrhagic episodes [152]. The management of hemophilia mainly depends on the replacement of the missing coagulation factor (episodic or requested treatment) or by preventing bleeding events [153]. The risk associated with repeated infusions of hematological products exposes hemophiliac patients to blood-transmitted disorders, including hepatitis or acquired immunodeficiency [154]. Therefore, cell and gene therapies offer durable production of coagulating factors and prevent patients from transmitted infections [153]. The next generation of recombinant products offers prolonged half-life, preventing frequent transfusions and, consequently, enabling superior quality of life [155]. Liver cell transplant has been demonstrated to be efficient and safe for FVIII deficiency but still exposes patients to risks associated with long-term immunosuppression and entails a waiting list for a compatible donor [116].

### 10.1. Hemophilia A

The first genetic approach for the treatment of hemophilia A (HA) was proposed almost 50 years ago. Currently, AAV and lentiviral (LV) vectors are preferred vectors for HA gene therapy. The potential application of LV-FVIII HSPC and liver-driven AAV-FVIII gene therapies to eliminate pre-existing inhibitors in hemophilia preclinical models and patients have been demonstrated [156]. Moreover, AAV5-hFVIII-SQ infusion was associated with the sustained normalization of coagulation factor for 1 year in almost all the participants [157,158]. Currently, several non-replacement agents are under investigation, including Fitusiran, Super FVa, factor Xa, APC inhibitors, and tissue factor pathway inhibitors (TFPI) [159,160]. Emicizumab (Hemlibra^®^; South San Francisco, CA, USA), a subcutaneously administered bispecific monoclonal antibody, has been recently approved, with or without inhibitors against FVIII. Such a monoclonal therapy stimulates the FVIII function but is also active in factor X and IX treatment [161]. The CRISPR-Cas9 technology has also been proposed for the treatment of HA. In a recent study, LNPs were preclinically used to deliver Cas9 mRNA along with single guide RNA (targeting AT), resulting in improvement in thrombin generation and reduction in bleeding-associated phenotypes [162]. In a phase 1/2 clinical trial (NCT04676048), the safety and preliminary efficacy of ASC618, an AAV vector encoding B-domain deleted codon-optimized human factor VIII under a synthetic liver-directed promoter, was investigated. Similar to NCT04676048, another phase 1/2 clinical trial uses a recombinant AAV vector composed of a bio-engineered capsid (AAV-Spark200) with liver-specific enhanced tropism and a codon-optimized expression cassette that encodes the SQ-FVIII variant of a B-domain-deleted human F8 gene. Preliminary results of SPK-8011 demonstrated that FVIII:C levels increased without exogenous factor infusions [163]. Moreover, in a phase 3 clinical trial, the Valoctocogene roxaparvovec (AAV5-hFVIII-SQ), an AAV5–based gene-therapy vector containing factor VIII complementary DNA, was used. The results demonstrated that application of Valoctocogene roxaparvovec provided endogenous factor VIII and significantly reduced bleeding time [27].

### 10.2. Hemophilia B

Currently, gene therapy efforts for the treatment of hemophilia B (HB) are mainly focused on the use of rAAV vectors that are systemically administered [164]. Moreover, gene-editing approaches using lipid nanoparticles to deliver mRNA encoding Cas9, gRNA, and a donor FIX cDNA template via a rAAV vector to knock in F9 and into the albumen locus have been used for the treatment of HB. The results demonstrated normal levels of FIX expression in an NHP model [165]. Moreover, lentiviral vectors are also used for systemic infusion in a mouse model [166]. As with HA, the CRISPR-Cas9 technology also has been proposed for the treatment of HB [167]. In a phase 1/2 clinical trial, the safety and efficacy of a single systemic administration of FLT180a in adult patients with HB has been investigated. The results showed durable FIX activity levels [168]. In a study sponsored by St Jude Children’s Research Hospital (SJCRH) and University College of London (UCL), AAV8-FIX-WT was infused to ten patients and the results demonstrated that sustained FIX activity up to 3–5% with no long-term safety concerns was achieved at year three [164]. Single-stranded AAV vector consisting of a bioengineered capsid, liver-specific promoter, and factor IX Padua (factor IX–R338L) was administered in ten men with hemophilia B who had factor IX coagulant activity of 2%. The new factor IX coagulant activity was sustained in all the participants, with a mean (±SD) steady-state activity of 33.7 ± 18.5%. Moreover, after 492 weeks follow-up, the annualized bleeding rate was significantly reduced [157].

## 11. Phenylketonuria

Phenylketonuria (PKU) is a common congenital metabolic disorder (1 in 10,000–14,000 live births) in which phenylalanine metabolism is defected due to allelic variations in the phenylalanine hydroxylase (*PAH*) gene. This gene is instrumental to convert phenylalanine to tyrosine [169,170]. In the absence of *PAH* activity, high levels of phenylalanine accumulate (360 µM) immediately after birth, resulting in deranged brain development and seizures and hypopigmentation of the skin, hair, and eyes [171]. PKU is the most common life-threatening single-gene Mendelian human disorder, whose treatment relies on dietary restrictions and key amino acid supplements, which delay, but do not erase, severe and sometimes permanent neurological dysfunction [171]. Currently, there is no cure for PKU, and dietary intervention results in grossly normal growth, with frequent gaps in IQ scores compared to unaffected siblings [172]. Discontinuation of dietary intervention in puberty and adulthood has been associated with regression in cognitive function, as well as adult-onset white matter degeneration, gait disturbances, and seizures [173]. Recently, liver and hepatocyte transplantation have been successfully offered by the University of Pittsburgh Medical Center (Pennsylvania, USA) [174]. Limits in life-long immunosuppression, limited number of donors, and short-term PAH activity have encouraged additional cell- and gene-based approaches. Eisensmith and colleague proposed somatic gene therapy using stable vectors [175], while another group reported efficacy in using AAV-mediated delivery of primary editing agents in preclinical models [176]. In a recent study, recombinant liver-tropical AAV2/8 vectors were used to deliver CRISPR-Cas9 to correct non-functional PAH by homologous recombination [177]. The non-homologous end-joining inhibitor, vanillin, was administered in combination with a viral vector to promote homologous repair, and the results showed lifelong, permanent correction of the PAH allele, leading to partial restoration of liver PAH activity [177]. An intravenous injection of AAV8-PAH resulted in long-term correction of hyperphenylalaninemia in male and female PKU mice [178], with no adverse hepatic events. Fibroblasts isolated from R408W guinea pigs demonstrated susceptibility to the correction using CRISPR or TALEN, with subsequent recombinant homology to *PAH* correction. The PKU guinea pig model provided a potent innovative platform for treatment of PKU and as a unique value for proof-of-concept studies for in vivo human gene-editing platforms [179]. In an intriguing recent study, a full-length mRNA encoding human PAH has been encapsulated in LNPs and delivered intravenously in PKU animal models, inducing high levels of human PAH enzyme and restoring its metabolism [180].

## 12. Maple Syrup Urine Disease

Maple syrup urine disease (MSUD) is a rare autosomal recessive heterogeneous metabolic disorder caused by a defect in the branched-chain alpha-ketoacid dehydrogenase (BCKD) complex, the second enzymatic step in the degradative pathway of the branched-chain amino acids (leucine, isoleucine, and valine) [181]. MSUD affects approximately 1 in 150,000 live births in the general population. However, in some isolated populations, such as the Mennonites, the prevalence is much higher [182]. MSUD is categorized as classic (severe), intermediate, or intermittent disorder and usually represents potentially lethal episodes of intoxication with acute neurological deterioration, nutritional problems, weight loss, and the smell of maple syrup in the urine [183]. Diet management increases the survival rate and reduces the risk of acute crises. However, mental and social impairments are still present in the majority of MSUD patients. Implementing a treatment regimen is challenging, and managing any metabolic crisis is uncertain and complex [184]. LT restores branched chain amino acids (BCAA) homeostasis but still faces short-term and long-term health risks [185]. Cell-based therapies (hepatocyte transplantation) have been demonstrated to be effective but are currently limited to preclinical studies [84]. Ectopic tissues, such as skeletal muscle, have been proposed as an alternative target for gene therapy. Greig and co-workers recently evaluated the efficacy of muscle and liver gene therapy [186]. AAV gene therapy based on the transfer of human BCKDHA or BCKDHB has been developed during the immediate neonatal period in MSUD animals [3]. The results supported BCKDHB gene transfer being successful (in recipient survival and normal growth) for 3 months, with significant improvement in the biochemical phenotype [3].

## 13. Progressive Familial Intrahepatic Cholestasis

Progressive familial intrahepatic cholestasis (PFIC) is a heterogeneous group of recessive hereditary liver diseases [187,188]. PFIC affects approximately 1 in 100,000 human beings worldwide [189]. Three different PFIC types with different mutations have been identified in the hepatocellular bile transport system: PFIC1, PFIC2, and PFIC3, which were caused by a mutation in *ATP8B1*, *ABCB11*, and *ABCB4*, respectively [187,190,191,192,193]. All types are present in infancy or childhood, leading to elevated serum bile salts, bilirubin, and pruritus. All forms of PFIC are associated with jaundice and elevated serum bile acid levels [187]. More than 30 different mutations have been observed in *ABCB4*, one-third of which are related to MDR3 expression [189]. The PFIC treatment option is ursodeoxycholic acid, which reduces the hydrophobicity of bile acid pools. The effectiveness of this treatment depends on the type of mutation. Patients with missense mutations respond better than patients with complete deficiency [194,195]. Most patients require LT in end-stage situations [196], while promising clinical outcomes have been reported by transplantation of a small amount of proficient hepatocytes [116]. Weber and colleagues treated PFIC mice with AAV vector expressing human *ABCB4* and successfully prevented PFIC3 symptoms in a clinically relevant mouse model [34]. Aronson and co-workers treated the same Abcb4^−/−^ mice with AAV8-hABCB4 recombinant vector and reported a reduction in hepatic damage and fibrosis and regeneration of bile phospholipid excretion [197].

## 14. Wilson Disease

Wilson disease (WD) is a rare autosomal recessive disorder caused by a mutation in the *ATP7B* gene. Epidemiologically, WD affects about 1 in 30,000 people worldwide [87]. Genetic defects lead to copper accumulation in the liver and eventually in the brain, associated with liver damage, and this could progress to neurological dysfunction in untreated individuals [198,199,200,201]. LT is the first-line treatment for fulminant liver failure, limited by availability of donor tissue. The success rate of WD surgery is significantly higher than other cases of acute liver failure. However, the mortality rate within one year after transplantation is high (10–20%) [202,203]. Recently, copper chelating agents have been approved (d’penicillamine, trientine, and zinc salts) and can stimulate enterocytes and allothioneine synthesis [204,205,206].

Hepatocyte transplantation in LEC rats as a model of WD has been performed. The results showed that cell transplantation eventually restored copper homeostasis and reversed liver disease in LEC rats [38]. The preclinical model developed for WD is *Atp7b* knock-out mice lacking *ATP7B* in the liver [207,208]. Murillo et al. treated *Atp7b* KO mice with AAV8-AAT-ATP7B vector, reporting correction of biochemical abnormalities, including high urinary copper excretion, low holoceruloplasminemia, high serum transaminase levels, and elevated hepatic copper levels [209]. Some clinical trials using AAVs are currently underway for the treatment of WD (NCT04884815, NCT04537377). A variety of heterozygous, homozygous, and compound heterozygous mutations and CRISPR-Cas9 technology could be used individually for each WD patient [210]. Pöhler and colleagues showed that CRISPR-Cas9 technology is efficient not only in introducing specific *ATP7B* mutations but also in correcting *ATP7B* point mutations [211]. Although the single-stranded oligo DNA nucleotide (ssODN) is limited to non-viral delivery methods, such application can lead to direct and safe modification in the point mutations within the *ATP7B* gene, providing therapeutic potential [212].

## 15. Glycogen Storage Diseases

Glycogen storage diseases (GSDs) are hereditary glycogen metabolic disorders [213]. GSDs manifest in abnormal amounts of glycogen in the liver, muscle, or both. There are 15 subtypes of GSDs generated by genetic defects in proteins involved in glycogen synthesis or degradation, glycolysis, and glucose release in the bloodstream [214,215]. The most severe types of GSDs are types I, III, and IV. The severity of the liver disease varies among these three types, but, ultimately, liver transplantation is the only treatment option [216]. Hepatocyte transplant has proven to be efficacious in mosaicism with proficient cells since the late 1990s [116]. Studies on targeted molecular therapies have yielded promising results, such as AAV carrying G6Pase [217]. In another study, a recombinant AAV vector containing a zinc finger nuclease (ZFN) targeted a site within the ROSA26 locus with no adverse effects [218]. The AAV-ZFN vector safely generates DNA breaks in the ROSA26 gene and enables the integration of the AAV G6Pase vector by homologous recombination. In the absence of ZFN, random cleavage of chromosomal DNA results in lower integration [219]. Landau and co-workers achieved stable G6Pase levels in the G6pc^−/−^ mouse model by integrating a human expression cassette at the ROSA26 as a safe harbor location of mice using ZFN [220].

GSD III, also known as Cori or Forbes disease, is a disorder in which glycogen breakdown is limited because of the defect in glycogen debrancher enzyme [91]. Excessive accumulation of the glycogen with short outer branches (limited dextrin) mainly in the liver and muscle is caused by a mutation in the *AGL* gene that causes a genetic deficiency of glycogen debranching enzyme (GDE) [221]. Although AAV gene therapy is a promising treatment for single-gene diseases, such as GSD III, the limited capacity of human GDE cDNA has led researchers to solve this problem by introducing a new gene therapy approach in GSD IIIa mice [222]. The authors used an AAV vector, which ubiquitously expressed a smaller bacterial GDE, Pullulanase, and intravenously injected the AAV vector (AAV9-CB-Pull) into the 2-week-old mice. GSD IIIa was tested and the results showed that it blocked glycogen accumulation in the heart and skeletal muscles but not in the liver, and it was associated with improved muscle function. Subsequent treatment with the liver-restricted AAV vector (AAV8-LSP-Pull) reduced liver glycogen content up to 75%. This approach reversed hepatic fibrosis, while it maintained the effect of AAV9-CB-Pull therapy on the heart and skeletal muscle. The results of this study indicated that AAV-mediated gene therapy with Pullulanase is a possible treatment approach for GSD III [222].

Recently, Pursell and colleagues investigated the LNP-mediated RNAi gene silencing approach. The results showed that glycogen synthase 2 was inhibited (preventing glycogen synthesis, glycogen accumulation, hepatomegaly, fibrosis, and the formation of liver nodules) in the GSD IIIa model [223]. GSD IV (also known as Anderson disease) is caused by abnormal glycogen accumulation because of the decreased activity of glycogen branching enzyme (GBE) [224]. Akman and colleagues developed a Gbe1^ys/ys^ mouse model [225] and Yi and co-workers used the recipient for AAV9 containing the GBE human expression cassette (AAV-GBE). Evaluation of the effectiveness of gene therapy in GSD IV showed that AAV treatment reduced damage and improved liver performance and muscle functions [226]. Another group evaluated CRISPR-Cas9 genome editing technology to correct a prevalent pathogenic human variant, G6PC-p.R83C. They treated newborn G6pc-R83C mice, with CRISPR-Cas9 reporting normal growth for 16 weeks without any hypoglycemic seizures [84].

## 16. Crigler–Najjar Syndrome

Crigler–Najjar syndrome [130] is a recessive hereditary metabolic and sporadic autosomal liver disorder. CN is characterized by severe unconjugated hyperbilirubinemia due to a marked decrease or complete deficiency in uridine diphospho glucuronosyl transferase 1A1 (*UGT1A1*) in the liver cells [227]. CN affects approximately 1 in 1,000,000 people worldwide [228]. Allogeneic hepatocytes transplantation has been demonstrated to be successful in the largest cohort of CN patients receiving such allogeneic cells so far [116], but it is limited in efficacy in the longer run. Thirty years later, hepatocyte transplants have been tested and validated in Gunn rats (a model for hyperbilirubinemia), and gene therapy has been tested on the same preclinical model. Although gene therapies via AAV vectors are promising approaches for the treatment of CN, not all patients are qualified for such approaches as the results of anti-AAV immunity occur due to previous exposure to the wild-type virus [229]. Greig and colleagues evolved an AAV8 vector expressing a codon-optimized human model of *UGT1A1* from a liver-particular promoter. High doses of the vector rescued neonatal lethality in newborn UGT1 KO mice, another model for CN syndrome, and appreciably extended the survival rate from 5 to 270 days [230]. In another study, *UGT1A1*-deficient mouse liver cell lines were generated to study the CN1 disease, and complete silencing of diacylglycerol acyltransferase-1 (DGAT1) was achieved by abrogating the entry of HCV in Huh-7.5 cells [43]. In another study, the researchers used TALEN technology to generate *UGT1A1*-deficient mice and Bortolussi and Muro tested the administration of a low-dose of AAV vectors in combination with SVP-rapamycin nanoparticles, reporting safety and efficacy in the long-term for CN [231].

## 17. Conclusions

Liver, as the central metabolic organ, is the target for cell and gene therapies as promising treatments for inherited and metabolic diseases. Gene-correction-based therapeutic modalities could help clinicians to effectively treat hereditary liver disorders. Currently, several gene therapy approaches, such as non-viral/viral-based vectors and CRISPR-Cas9 genome editing technology, have provided long-lasting therapeutic effects in clinically relevant animal models. The efficacy of viral vectors for gene therapy was confirmed, but their practical application faces several limitations. Since the presence of viral genetic material in the plasmid can induce an acute immune response and a possible oncogenic transformation, there are strong aggravating factors for broad application of them. On the other hand, CRISPR-Cas9 genome editing technology may cleave DNA at the target site, which is possibly limiting the application of Cas9 proteins or leading to harmful effects. Thus far, few gene therapy approaches have been successfully translated to clinical studies. Ongoing clinical trials encourage such approaches and re-invigorate optimization for liver monogenic and metabolic disorders. Although many challenges remain to be addressed, ongoing efforts and recent promising results support researchers in refining and enhancing gene-based therapy approaches for the treatment of complicated genetic disorders. 

## Figures and Tables

**Figure 1 bioengineering-09-00392-f001:**
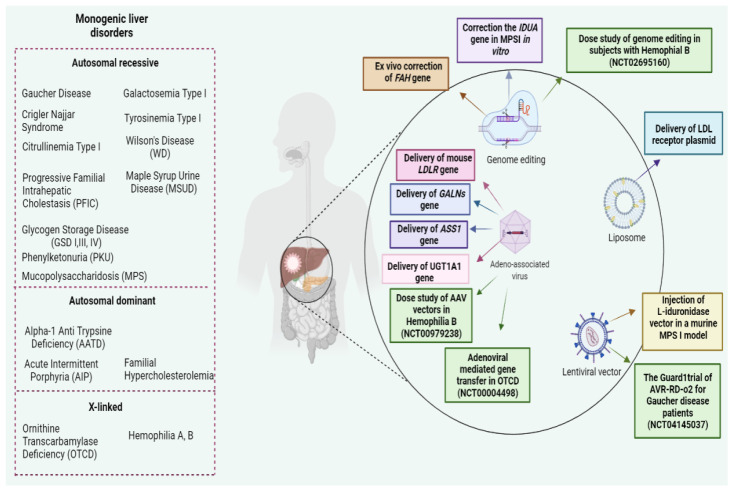
Common monogenic liver disorders and proposed gene therapy approaches. The figure was created with Biorender (www.biorender.com (accessed on 21 June 2022)).

**Figure 2 bioengineering-09-00392-f002:**
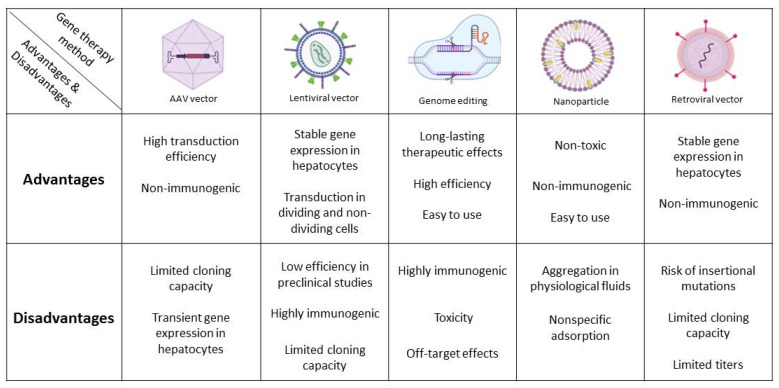
The advantages and disadvantages of common gene delivery methods in monogenic liver disorders. The figure was created with Biorender (www.biorender.com (accessed on 21 June 2022)).

**Table 1 bioengineering-09-00392-t001:** Therapeutic options for liver genetic disorders: liver transplant, hepatocyte transplant, and gene therapy.

Genetic Disorders	Liver Transplant	Hepatocyte Transplant	Gene Therapy
1. Familial Hypercholesterolemia	Clinical study, W. Bilheimer et al. [8]	Clinical study, Grossman et al. [9]	Clinical study, (NCT02651675)
2. Gaucher Disease	Clinical study, M. Ayto et al. [10]	N.A.	Clinical study, (NCT05139316)
3. Mucopolysaccharidosis	Preclinical study, Toyama et al. [11]	N.A.	Clinical study, (NCT04201405)
4. Urea cycle defects			
a. OTC Deficiency	Clinical study, A. Busuttil et al. [12]	Clinical study, Stéphenne et al. [13]	Clinical study, (NCT05092685)
b. Citrullinemia type I	Clinical study, Yuan et al. [14]	Clinical study, Meyburg et al. [15]	Preclinical study, Chandler et al. [16]
5. Alpha-1-anti Trypsin Deficiency	Clinical study, Hood et al. [17]	Preclinical study, Kay et al. [18]	Clinical study, (NCT04474197)
6. Tyrosinemia Type I	Clinical study, Freese et al. [19]	Clinical study, Ribes-Koninckx et al. [20]	Preclinical study, VanLith et al. [21]
7. Galactosemia	Clinical study, Otto et al. [22]	N.A.	Preclinical study, Rasmussen et al. [23]
8. Acute Intermittent Porphyria	Clinical study, F. Soonawalla et al. [24]	N.A.	Clinical study, (NCT02082860)
9. Hemophilia	Clinical study, Kurian et al. [25]	Clinical study, Kohei et al. [26]	Clinical study, Ozelo et al. [27]
10. Phenylketonuria	Clinical study, Vajro et al. [28]	Clinical study, Stéphenne et al. [29]	Clinical study, (NCT04480567)
11. Maple Syrup Urine Disease	Clinical study, Wendel et al. [30]	Preclinical study, Skvorak et al. [31]	Clinical study, (NCT03173521)
12. Progressive familial intrahepatic cholestasis	Clinical study, Aydogdu et al. [32]	Preclinical study, De Vree et al. [33]	Preclinical study, Weber et al. [34]
13. Wilson Disease	Clinical study, Bellary et al. [35]	Preclinical study, Allen et al. [36]	Clinical study, (NCT04884815)
14. Glycogen Storage Diseases	Clinical study, Li et al. [37]	Preclinical study, Malhi et al. [38]	Clinical study, (NCT00976352)
15. Crigler–Najjar Syndrome	Clinical study, Rela et al. [39]	Clinical study, Ambrosino et al. [40]	Clinical study, (NCT03466463)

**Table 2 bioengineering-09-00392-t002:** Gene-therapy-based clinical trials in monogenic liver disorders.

Hereditary Disease (Monogenic Liver Disorder)	Gene Therapy Approach	Status	Phase	Outcome of Intervention	NCT Number
Ornithine Transcarbamylase Deficiency (OTCD)	single dose of recombinant adenovirus infused into the liver under fluoroscopic guidance	Terminated	Phase 1	Not Provided	NCT00004386
	Intravascular adenoviral vector mediated gene transfer into the live	Terminated	Phase 1	Not Provided	NCT00004498
	HORACE ^1^ (AAVLK03hOTC); specifically targets the liver	Not yet recruiting	Phase 1/2	Efficacy and safety outcomes	NCT05092685
	AAV serotype 8 (AAV8)-Mediated Gene Transfer	Recruiting	Phase 3	Change in plasma ammonia (AUC0-24) from baseline to week 64 for all participants	NCT05345171
	single IV infusion of DTX301 (scAAV8OTC)	Completed	Phase 1/2	Change in baseline in ureagenesis rate	NCT02991144
Maple Syrup Urine Disease (MSUD)	AAV8 for the delivery of the human ARSB gene (AAV2/8.TBG.hARSB ^2^) to liver	Active, not recruiting	Phase 1/2	Efficacy outcome	NCT03173521
Phenylketonoria (PKU)	single I.V. administration AAVHSC15 vector containing a functional copy of the human *PAH* gene (HMI-102)	Recruiting	Phase 1/2	Change in plasma Phe concentration from baseline	NCT03952156
	AAV-mediated gene transfer of BMN 307	Active, not recruiting	Phase 1/2	Change from baseline in mean plasma Phe levels	NCT04480567
	IV administration of HMI-103 AAVHSC15 vector containing a functional copy of the human *PAH* gene	Active, not recruiting	Phase 1	Change from baseline in natural and total protein intake (g/day) at each timepoint post-administration of HMI-103	NCT05222178
Alpha-1-anti Trypsin Deficiency (AATD)	Oral administration of VX-864 iRNA	Completed	Phase 2	Change in plasma antigenic AAT levels	NCT04474197
	Administration of a serotype rh.10 replication deficient AAV expressing the human alpha-1 antitrypsin cDNA (ADVM-043)	Completed	Phase 1/2	Change in therapeutic serum and alveolar epithelial lining fluid levels of a1AT as a preliminary measure of efficacy	NCT02168686
	rAAV2-CB-hAAT gene Vector	Completed	Early Phase 1	Human AAT levels and phenotype in the blood	NCT00377416
	rAAV1-CB-hAAT	Completed	Phase 1	Human AAT levels and phenotype in the blood	NCT00430768
Acute Intermittent Porphyria (AIP)	rAAV2/5-PBGD	Completed	Phase 1	Health-related quality of life of AIP patients	NCT02082860
	Gene therapy rAAV2/5-PBGD for the treatment of acute intermittent porphyria	Completed	Phase 1	Effect of the treatment on porphobilinogen (PBG) and delta-aminolevulinic acid (ALA) urinary level.	NCT02082860
Hemophilia B	IV infusion of SPK-9001 ^3^	Completed	Phase 2	Change from baseline in FIX:C Antigen Level at Steady State	NCT02484092
	Genome editing by zinc finger nuclease therapeutic SB-FIX	Terminated	Phase 1	Effect of SB-FIX on presence and shedding in AAV2/6 vector DNA	NCT02695160
	AAV-mediated gene transfer of scAAV2/8-LP1-hFIXco	Active	Phase 1	Not Provided	NCT00979238
	Using a Single-Stranded, Adeno-Associated Pseudotype 8 Viral Vector (AAV8-hFIX19)	Terminated	Phase 1	Factor IX activity and antigen; PT; and aPTT.	NCT01620801
	AAV vector containing Factor IX gene named FLT180a	Terminated	Phase 1/2	Change from baseline in FIX concentrate consumption and annualized bleeding rate	NCT03369444
	AAV containing BBM-H901 ^4^	Active, not recruiting	Not applicable	Vector-derived FIX:C and FIX antigen levels.	NCT04135300
Hemophilia A	single IV infusion of ASC618 ^5^	Not yet recruiting	Phase 1/2	Changes in FVIII activity levels from baseline	NCT04676048
	novel AAV vector (with a stronger attraction to the human liver) to deliver the human factor VIII (hFVIII) named SPK-8011	Recruiting	Phase 1/2	Increased FVIII:C levels to prevent spontaneous bleeding	NCT03003533
	AAV-based gene therapy (Valoctocogene roxaparvovec ^6^)	Active, not recruiting	Phase 1/2	Frequency of FVIII replacement therapy during the study	NCT02576795
	Infusion of AAV2/8-HLP-FVIII-V3	Recruiting	Phase 1	Plasma hFVIII activity	NCT03001830
Mucopolysaccharidosis	Autologous CD34^+^ cells transduced with a lentiviral vector containing the human N-Sulfoglucosamine Sulfohydrolase (*SGSH*) gene	Active, not recruiting	Phase 1/2	change in ng/mL glycosaminoglycans in CSF from baseline following IMP administration	NCT04201405
	Retroviral-mediated gene transfer of Lymphocyte gene	Completed	Phase 1/2	Not Provided	NCT00004454
	Genome editing by zinc finger nuclease for SB-318	Terminated	Phase 1/2	Effect of SB-318 on leukocyte IDUA activity	NCT02702115
Fabry Disease Lysosomal Storage Diseases	Single-ascending dose study of a novel AAV containing FLT190	Recruiting	Phase 1/2	Frequency of treatment-emergent adverse events (AEs)	NCT04040049
	Single dose of investigational product, ST-920 ^7^	Recruiting	Phase 1/2	Incidence of treatment-emergent adverse events (TEAEs)	NCT04046224
Wilson Disease	AAV-mediated gene transfer using infusion of UX701	Recruiting	Phase 1/2	Change in Liver Copper Concentration	NCT04884815
	Recombinant AAV-mediated gene transfer of VTX-801	Recruiting	Phase 1/2	Serum ceruloplasmin activity (enzymatic assay)	NCT04537377
Familial Hypercholesterolemia	Low Density Lipoprotein Receptor mRNA Exosomes	Not yet recruiting	Phase 1	Changes in Stability of Carotid Artery Plaques	NCT05043181
	Recombinant retroviral vector (ex-vivo liver directed gene therapy)	Completed	Phase 1	Not Provided	NCT00004809
	AAV directed hlDLR gene therapy	Completed	Phase 1/2	Percent change in LDL-C compared to baseline	NCT02651675
Gaucher disease	Lentiviral-mediated gene transfer of AVR-RD-02	Recruiting	Phase 1/2	Change from Baseline in plasma Chitotriosidase activity levels	NCT04145037
	Retroviral-mediated gene transfer containing human glucocerebrosidase cDNA (ex vivo)	Completed	Phase 1	Not Provided	NCT00001234
Glycogen Storage Diseases	AAV8-mediated gene transfer of DTX401	Recruiting	Phase 3	Change from Baseline to Week 48 in Time to Hypoglycemia	NCT05139316
	Recombinant AAV1-mediated gene transfer of rAAV1-CMV-GAA	Completed	Phase 1/2	Change in AAV antibody level; change in Alglucosidase alpha (GAA) Antibody level; maximal inspiratory pressure	NCT00976352
	AAV8-mediated gene transfer of AT845	Recruiting	Phase 1/2	Change from baseline in thigh fat fraction	NCT04174105
Crigler–Najjar Syndrome	AAV-mediated gene transfer of GNT0003	Recruiting	Phase 1/2	Decrease in total Serum bilirubin level	NCT03466463

^1^ Halting Ornithine Transcarbamylase Deficiency with Recombinant AAV in ChildrEn (HORACE). ^2^ Arylsulfatase B. ^3^ Adeno-associated Viral Vector with Human Factor IX Gene. ^4^ An adeno-associated viral (AAV) vector designed to drive expression of the human factor IX (hFIX). ^5^ AAV vector encoding B-domain deleted codon-optimized human factor VIII under a synthetic liver-directed promoter. ^6^ Adenovirus-Associated Virus Vector-Mediated Gene Transfer of Human Factor VIII. ^7^ Recombinant AAV2/6 vector encoding the cDNA for human a-Gal A.

## Data Availability

All data supporting the findings of this study are available within the article and its supplemental data file or from the corresponding author upon reasonable request.

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
