# Peer review of "Novel Gene-Correction-Based Therapeutic Modalities for Monogenic Liver Disorders"

_bioengineering, 2022, doi:10.3390/bioengineering9080392_

Round 1
Reviewer 1 Report
A better description of the clinical trials summarized in table 2 should be added in the text because clinical trials are the source of the most relevant data.
There are a lot of studies missing like the one sing siRNA or LNPs-mRNA.
The whole text and tables should be carefully reviewed and appropriate and up to date references used avoiding the use of reviews.
When an abbreviation has been described it should be used in the text for example when refereeing to AAV vectors.
Line 71 “Studies have proved as recombinant AAV” substitute by “Studies have proved that recombinant AAV”
Line 77: This sentence has to be modified “Non-viral vectors are synthesized in laboratories and proposed due to their superior safety and relatively easy-to-construct features compared to viral vectors”. Proposed for what? proposed as an alternative to viral vectors?
Line 80: “Cellular extra vesicles” substitute by “extracellular vesicles”
Line 83: “Beside to delivering vectors” “Beside delivering vectors”
Line 102 Rewrite the sentence: Furthermore, the administration of potent statins like atorvastatin have shown reduced plasma LDL-C with an efficiency in only 10%-25% of cases [15].
Line 116 mice FH model “FH mouse model
Line 117 “The results supported this intervention to retain LDLR expression and decrease the atherosclerosis and LDL cholesterol levels” “The results showed that this intervention resulted in LDLR expression and decrease the atherosclerosis and LDL cholesterol levels”
Gaucher disease: indicate that it is mainly due to the accumulation of glucocerebroside in macrophages that affect many organs.
Substitute GC gene by GBA gene or use GCase when referring to Glucocerebrosidase enzyme
Line 139 “Moreover, researchers transferred a rAAV vector containing GC cDNA into the fibroblasts and fetal kidney cells derived from GD patients” this is not correct they transfer the cDNA to patients fibroblast and to HEK 293 cells that are embryo kidney fibroblast but not from patients
Line 141 “expression was reported sustained for 12 and 20 weeks, respectively in vitro and in vivo” “sustained expression was reported for 12 and 20 weeks, in vitro and in vivo, respectively”
More recent publications have to be cited like:
Massaro G, Hughes MP, Whaler SM, Wallom KL, Priestman DA, Platt FM, Waddington SN, Rahim AA. Systemic AAV9 gene therapy using the synapsin I promoter rescues a mouse model of neuronopathic Gaucher disease but with limited cross-correction potential to astrocytes. Hum Mol Genet. 2020 Jul 29;29(12):1933-1949. doi: 10.1093/hmg/ddz317. PMID: 31919491; PMCID: PMC7390934.
Muchopolisaccharidosis. The clinical trials for Hunter and Hurler performed using ZFN should be mention
Line 189, adenovirus associated (AAV) substitute by AAV.
Line 196 OTD deficiency by OTC deficiency
Reference 42 is not correct.
Prieve MG, Harvie P, Monahan SD, Roy D, Li AG, Blevins TL, Paschal AE, Waldheim M, Bell EC, Galperin A, Ella-Menye JR, Houston ME. Targeted mRNA Therapy for Ornithine Transcarbamylase Deficiency. Mol Ther. 2018 Mar 7;26(3):801-813. doi: 10.1016/j.ymthe.2017.12.024. Epub 2018 Jan 4. PMID: 29433939; PMCID: PMC5910669.
Line 239: use AAV
Line 243: reference 42 it is not the correct one
Explain what is AAT-Z and AAT-M
Line 248: This sentence “A critical challenge in efficient delivery of the CRISPR-Cas9 system is the limited size of viral vector transgenes, particularly in AAV vectors” has not sense in this paragraph it should be mentioned when talking about AAV as vector, and what is limited is the cloning capacity of the viral genome no the vector transgenes.
Line 253: what is AAT 8/ AVL.
Line 308. Balakrishnan and colleagues reported that intravenous injection of human GALT mRNA into deficient mice was effective in inducing hepatic expression and long-term enzyme activity, and significantly reducing plasma galactose concentration” It should be indicated that the mRNA is delivered using LNPs and that several dos have to be administered to achieve long term expression since the expression from LNPs-mRNA is transient.
Line 334. In the clinical trial by D’avola the efficacy of AAV5 was not compared with AAV8.
AAV5 and AAV8 were used to perform preclinical studies in AIP mouse model
When referring to the use of LNPs carrying the PBGD mRNA the paper by should be cited and not a review by the author’s
Jiang L, Berraondo P, Jericó D, Guey LT, Sampedro A, Frassetto A, Benenato KE, Burke K, Santamaría E, Alegre M, Pejenaute Á, Kalariya M, Butcher W, Park JS, Zhu X, Sabnis S, Kumarasinghe ES, Salerno T, Kenney M, Lukacs CM, Ávila MA, Martini PGV, Fontanellas A. Systemic messenger RNA as an etiological treatment for acute intermittent porphyria. Nat Med. 2018 Dec;24(12):1899-1909. doi: 10.1038/s41591-018-0199-z. Epub 2018 Oct 8. PMID: 30297912.
The hemophilia paragraph should be rewritten introducing first the different therapeutic alternatives and finishing with the gene therapy alternatives, an in particular with the clinical trials with AAV for hemophilia A and B that are very close to marketing authorization.
Line 403. Correct AAV8-PAL by AAV8-PAH
The fact that clinicals trials using AAV are ongoing for Wilson disease should be mentioned in the text.
Line 488, no slower but lower integration.
The reference 183 does not appear in pubmed neither a reference on the use of SVP-Rapamycin for Cigler Najjar.
Table 1: I think a more appropriate title will be therapeutic option for liver genetic disorders: liver transplant, hepatocyte transplant and Gene therapy. The description about what is label in red or green is not accurate.
Hepatocyte transplant has been performed in Wilson disease using LEC rats.
Figure 2. One of the advantages of nanoparticles indicated by the authors is “shorter incubation time” it is not clear for this reviewer
According to the authors one of the disadvantages of AAV is transient gene in expression in hepatocytes, but it is just the opposite AAV are characterized by long term expression. The expression is transient in neonates, newborns in which the hepatocytes divide and the AAV genome is lost to its episomal nature. Transient gene expression is a limitation of non-viral vectors.
Table 2: NCT02076763 is an observational study in AIP patients the AAV was not administered It appear twice in the table.
Studies that are not interventional have been included and should be removed.
The authors should reorganize the table because AATD trials are in the AIP section and viceversa.
Author Response
Dear Reviewer 1,
Bioengineering
Thank you for giving the opportunity to re-submit the manuscript entitled, “Novel Gene-Correction based Therapeutic Modalities for Monogenic Liver Disorders [ID: 1806108].” to be considered for publication in Bioengineering. The comments were helpful and improved the quality of manuscript. We have modified the manuscript accordingly and have added all the necessary explanations to the article based on the points raised by the reviewer. The new and modified version of the manuscript marked up using the “Track Changes” function. We hope this update make this manuscript more useful for your readers.
We look forward to hearing from you.
Sincerely,
Massoud Vosough MD., Ph.D.
Head of Regenerative Medicine Department
Royan Institute for Stem Cell Biology and Regenerative Medicine
Email address: masvos@royaninstitute.org

Reviewer 2 Report
The review entitled “Novel Gene-Correction based Therapeutic Modalities for Monogenic Liver Disorders” by Ghasemzad and co-authors provides a broad overview of gene therapy and genome editing technologies targeting liver disorders.
While this is a reasonably comprehensive overview, there are a number of issues that need to be addressed. In addition, the manuscript could benefit from editing to improve the use of English and flow of the paper.
Specific comments for the authors:
1. The title does not fully capture the content of the review rather implies that the review will cover gene correction; gene addition and other approaches are also discussed.
2. Line 38: the statement that gene editing technology is providing the “most promising” results
3. Line 39: “CrispR “should be “CRISPR” and should also be described in full on line 84 which is the first time this is used in the body of the text.
4. Line 47, line 55 and throughout. The word “the” could be used more often. For example, I would say “The liver is a key metabolic organ”.
5. Overall, the standard of English could be improved and there are many sentences that don’t make sense. For example on line 68, the statement “Viral vectors are important gene transferring mediator” is not complete. Indeed, this whole paragraph seems to move from one idea to another quickly without any context.
6. Again, line 83, this statement does not make sense “Beside to delivering vectors, genome editing tools such as…” - genome editing tools do not deliver vectors. The editing tools are the technology, not the delivery system.
7. Line 183, OTC is a mitochondrial enzyme, not a cytoplasmic enzyme.
8. Section on OTC. Again, this section jumps from editing to LNP to AAV gene addition without introducing the concepts making it disjointed. I would also suggest including reference to the work of Alexander who has published extensively in OTC gene therapy.
9. Line 484, reference 168 doesn’t seem to be the correct reference. I would suggest cross-checking the references.
10. Table 1. The header for this table is “Cell and organ transplantation…” but there is also a gene therapy column. It is also confusing that the red boxes are in the gene therapy section but the header says that red refers to “transplantation performed in preclinical studies”?
11. Figure 2. The figure describes “gene delivery methods” but again genome editing is included here. Genome editing is the technology and can be delivered but any of the vector systems. Also one advantage of AAV is listed as “safe” because there are more clinical trials. Firstly, there have been more trials using retroviral and adenoviral vectors (see https://a873679.fmphost.com/fmi/webd/GTCT) and just because there are “more” trials doesn’t make something safer.
12. Conclusion – again very brief and poorly written. For example the authors state that “Although many challenges have been remained to address, ongoing efforts and promising results support researchers in refine and enhance gene-based therapy approaches for the treatment of complicated genetic disorders.” It would be good to state what some of these challenges are and what are the implications/how could approaches be refined etc.?
Author Response
Dear Reviewer 2,
Bioengineering
Thank you for giving the opportunity to re-submit the manuscript entitled, “Novel Gene-Correction based Therapeutic Modalities for Monogenic Liver Disorders [ID: 1806108].” to be considered for publication in Bioengineering. The comments were helpful and improved the quality of manuscript. We have modified the manuscript accordingly and have added all the necessary explanations to the article based on the points raised by the reviewer. The new and modified version of the manuscript marked up using the “Track Changes” function. We hope this update make this manuscript more useful for your readers.
We look forward to hearing from you.
Sincerely,
Massoud Vosough MD., Ph.D.
Head of Regenerative Medicine Department
Royan Institute for Stem Cell Biology and Regenerative Medicine
Email address: masvos@royaninstitute.org

Reviewer 3 Report
The authors review the main monogenic liver diseases and describe the different gene therapy approaches developed.
The document is clear, well written and very enjoyable to read.
Only some points need to be clarified/added.
Major point:
In the introduction (p.2, l.65), the authors mention the two main approaches developed, i.e. in vivo or ex vivo, with the use of hepatocytes isolated from patients. However, the use of primary cells or even differentiated cells from stem cells (cell and gene therapy) is not mentioned and should be added in each section.
Ex: For FH, no mention is made of studies performed on human cells (primary or derived from stem cells) (Choisat 2015; Omer 2021; Caron 2019;...); in the section on Gaucher disease, no mention has been made concerning the use of human induced pluripotent stem cells (Panicker LM 2012; Zhao Q, 2020;...) that could be corrected and differentiated into the cell type of interest. The recent mouse model of Guo et al should also be mentioned; for AATD, targeted gene correction in PSC has also been performed (Yusa 2011, Rashid 2012,...); and so on.
In the hemophilia part, HA and HB should be divided in two subparts in order to be clearer and develop better the HB part that has been poorly evocated with ex vivo (Luce 2022;...) and in vivo parts (Nathwani 2011,...).
Minor points:
- The sentence "non-viral vectors are synthesized in laboratories and proposed due to their superior safety and relatively easy-to-construct features compared to viral vectors" (p.2, l.77) should be qualified as some of them are much more difficult to develop than "classical" viral vectors.
- Please add retro and adenorial vectors in figure 2 as you mentioned it in the main text.
Author Response
Dear Reviewer 3,
Bioengineering
Thank you for giving the opportunity to re-submit the manuscript entitled, “Novel Gene-Correction based Therapeutic Modalities for Monogenic Liver Disorders [ID: 1806108].” to be considered for publication in Bioengineering. The comments were helpful and improved the quality of manuscript. We have modified the manuscript accordingly and have added all the necessary explanations to the article based on the points raised by the reviewer. The new and modified version of the manuscript marked up using the “Track Changes” function. We hope this update make this manuscript more useful for your readers.
We look forward to hearing from you.
Sincerely,
Massoud Vosough MD., Ph.D.
Head of Regenerative Medicine Department
Royan Institute for Stem Cell Biology and Regenerative Medicine
Email address: masvos@royaninstitute.org

Reviewer 4 Report
The manuscript “Novel Gene-Correction based Therapeutic Modalities for Monnogenic Liver Disorders” has summarized the common monogenic liver diseases and their associated gene therapy treatment strategies. The author particularly focused on CRISPR-Cas9 technology and provided a nice summary of recent works. Before publication, there are a few comments needed to be addressed.
First, the authors need to do a format and language check of the whole manuscript. Some mistakes are not acceptable. For example, line 39, it should be CRISPR-Cas9. Similarly line 86. Make sure all the gene names are italic and the mice model names are correct with lower case and italic too. For example, line 113, the mice model of LDLR is not correct. Make sure to italicize “in vitro” “in vivo” and “ex vivo”. And it should be Crigler-Najjar syndrome in line 522.
Line 90, I would say it is better to provide a section of clinical progress of the gene correction-based therapies. Table 2 did summarize the work, but the information there is very limited and can be confusing, especially outcomes. It will be nice to provide a section to summarize and highlight the good works, along with table 2.
Section 6, the authors need to make sure to use one expression of Alpha-1-anti Trypsin (Alpha-1-antitrypsin)
For figure2, I think the authors need to spend more effort in providing the advantages and disadvantages of each gene delivery method. First of all, I don’t think genome editing is a delivery method in the first place. A lot of the advantages and disadvantages are not true. For example, AAV vector is sate is an overstatement, because a lot of the clinical trials were terminated due to patient complications. For lentiviral vectors, there are apparently more disadvantages than just low efficiency. And for nanoparticles, few clinical studies cannot be a disadvantages.
Overall, the manuscript is a good summary but needs more efforts towards the depth.
Author Response
Dear Reviewer 4,
Bioengineering
Thank you for giving the opportunity to re-submit the manuscript entitled, “Novel Gene-Correction based Therapeutic Modalities for Monogenic Liver Disorders [ID: 1806108].” to be considered for publication in Bioengineering. The comments were helpful and improved the quality of manuscript. We have modified the manuscript accordingly and have added all the necessary explanations to the article based on the points raised by the reviewer. The new and modified version of the manuscript marked up using the “Track Changes” function. We hope this update make this manuscript more useful for your readers.
We look forward to hearing from you.
Sincerely,
Massoud Vosough MD., Ph.D.
Head of Regenerative Medicine Department
Royan Institute for Stem Cell Biology and Regenerative Medicine
Email address: masvos@royaninstitute.org

Round 2
Reviewer 1 Report
A better description of the AAV clinical trials for Hemophilia B must be included.
The clinical trial table should be reviewed: 1. AIP clinical is in the AATD paragraph. 2. Non-intervantional trial are include and shoul be removed (NCT03520712, NCT04909346) 3. For Hemophilia A and B several trials are missing. 4. In the gene therapy approach column AAV-mediated gene transfer of UX701 for example it is not correct since UX701 is the name of the AAV vector, replace by infusion of UX701 or AAV9-mediated gene transfer of ATP7B.
References:
reference 1 and 3 is the same publication.
Journal information is missing in many references: 9, 10, 13, 15-21.........
Author Response
Re: Manuscript ID: 1806108
Dear Reviewer 1,
Bioengineering
Thank you for giving us the opportunity to re-submit the manuscript entitled, “Novel Gene-Correction, -Editing based Therapeutic Modalities for Monogenic Liver Disorders [ID: 1806108].” to be considered for publication in Bioengineering. The 2nd round of reviewer’s comments were helpful and improved the quality of manuscript. We have modified the manuscript accordingly and have added all the necessary explanations to the article based on the points raised by the reviewer. The new and modified version of the manuscript marked up using the “Track Changes” function. We hope this update make this manuscript more useful for your readers.
We look forward to hearing from you.
Sincerely,
Massoud Vosough MD., Ph.D.
Head of Regenerative Medicine Department
Royan Institute for Stem Cell Biology and Regenerative Medicine
Email address: masvos@royaninstitute.org

Reviewer 2 Report
The revised version of the manuscript submitted by Ghasemzad and co-workers addresses the concerns of the reviewers.
Author Response
Re: Manuscript ID: 1806108
Dear Reviewer 2,
Bioengineering
Thank you for reviewing the manuscript entitled, “Novel Gene-Correction, -Editing based Therapeutic Modalities for Monogenic Liver Disorders [ID: 1806108].” to be considered for publication in Bioengineering.
Sincerely,
Massoud Vosough MD., Ph.D.
Head of Regenerative Medicine Department
Royan Institute for Stem Cell Biology and Regenerative Medicine
Email address: masvos@royaninstitute.org
Reviewer 3 Report
The authors have taken into account most of the comments and corrections requested after the first reading of the article. The review is almost complete and pleasant to read.
Major comment:
Please add iPSC derivation and correction (+ appropriate references) for mucopolysaccharidosis, Ornithine Transcarbamylase Deficiency, citrullinemial type I, alpha 1 anti-trypsin, tyrosinemia type I, hemophilia A and B,... as numerous studies have been performed using ex vivo editing based therapeutic approaches using iPSCs that did not mention in the current version of the review
Minor comments:
- choose between CRISPR/Cas9 and CRISPR-Cas9
- p13 l. 620 Crigler-Najjar and not Najjer
Author Response
Re: Manuscript ID: 1806108
Dear Reviewer 3,
Bioengineering
Thank you for giving us the opportunity to re-submit the manuscript entitled, “Novel Gene-Correction, -Editing based Therapeutic Modalities for Monogenic Liver Disorders [ID: 1806108].” to be considered for publication in Bioengineering. The 2nd round of reviewer’s comments were helpful and improved the quality of manuscript. We have modified the manuscript accordingly and have added all the necessary explanations to the article based on the points raised by the reviewer. The new and modified version of the manuscript marked up using the “Track Changes” function. We hope this update make this manuscript more useful for your readers.
We look forward to hearing from you.
Sincerely,
Massoud Vosough MD., Ph.D.
Head of Regenerative Medicine Department
Royan Institute for Stem Cell Biology and Regenerative Medicine
Email address: masvos@royaninstitute.org

Reviewer 4 Report
I have no more comments. Thanks!
Author Response
Re: Manuscript ID: 1806108
Dear Reviewer 4,
Bioengineering
Thank you for reviewing the manuscript entitled, “Novel Gene-Correction, -Editing based Therapeutic Modalities for Monogenic Liver Disorders [ID: 1806108].” to be considered for publication in Bioengineering.
Sincerely,
Massoud Vosough MD., Ph.D.
Head of Regenerative Medicine Department
Royan Institute for Stem Cell Biology and Regenerative Medicine
Email address: masvos@royaninstitute.org